# Renal Rehabilitation: Exercise Intervention and Nutritional Support in Dialysis Patients

**DOI:** 10.3390/nu13051444

**Published:** 2021-04-24

**Authors:** Junichi Hoshino

**Affiliations:** Nephrology Center, Toranomon Hospital, Tokyo 105-8470, Japan; hoshino@toranomon.gr.jp; Tel.: +81-3-3588-1111

**Keywords:** renal rehabilitation, exercise, sarcopenia and frailty, nutritional support, protein synthesis, muscle physiology, dialysis, physical activity, exercise tolerance, quality of life

## Abstract

With the growing number of dialysis patients with frailty, the concept of renal rehabilitation, including exercise intervention and nutrition programs for patients with chronic kidney disease (CKD), has become popular recently. Renal rehabilitation is a comprehensive multidisciplinary program for CKD patients that is led by doctors, rehabilitation therapists, diet nutritionists, nursing specialists, social workers, pharmacists, and therapists. Many observational studies have observed better outcomes in CKD patients with more physical activity. Furthermore, recent systematic reviews have shown the beneficial effects of exercise intervention on exercise tolerance, physical ability, and quality of life in dialysis patients, though the beneficial effect on overall mortality remains unclear. Nutritional support is also fundamental to renal rehabilitation. There are various causes of skeletal muscle loss in CKD patients. To prevent muscle protein catabolism, in addition to exercise, a sufficient supply of energy, including carbohydrates, protein, iron, and vitamins, is needed. Because of decreased digestive function and energy loss due to dialysis treatment, dialysis patients are recommended to ingest 1.2-fold more protein than the regular population. Motivating patients to join in activities is also an important part of renal rehabilitation. It is essential for us to recognize the importance of renal rehabilitation to maximize patient satisfaction.

## 1. Introduction

An aging society is a worldwide problem, especially in developed countries. For example, more than 28% of Japan’s population was over 65 years old in 2018, the highest proportion in the world, and by 2030, one-third of the population will be 65 or older and one-fifth will be 75 or older. One of the main reasons for this phenomenon in the developed countries is the prolonged life expectancy at birth. Fifty years ago, life expectancy at birth in Japan was approximately 72 years, but it has since climbed to 84 years. On the contrary, prolonged life expectancy is not equivalent to a healthier life expectancy at birth (HALE), which is defined by the World Health Organization as the average number of years of full health that a newborn could expect to live if he or she were to pass through life subject to the age-specific death rates and ill-health rates of a given newborn. It has been reported that the difference between HALE and life expectancy is approximately nine years in males and 12 years in females [1]. The difference could be considered as equal to the number of years of life during which one needs the support of nursing and family care, which patients and healthcare providers should work together to minimize.

This phenomenon has also been observed in chronic kidney disease (CKD) patients, including dialysis patients. As patients with the end stage renal disease (ESRD) rates tend to rise with age, the 2018 annual data report by the United States Renal Data System revealed that trends in the incidence rate of treated ESRD patients from 2003 to 2016 have remained high in approximately half of countries [2]. The Japanese Society for Dialysis Therapy Renal Data Registry (JRDR) reported that the number of chronic dialysis patients in Japan has been increasing every year, and reached more than 339,841 patients (the prevalence ratio was 2688 per million population) with the mean age of all dialysis patients was 68.8 years at the end of 2018 [3]. Thanks to great advances in dialysis technologies, the circumstances of dialysis patients have been changing dramatically. For example, the proportion of facilities with ultrapure dialysate (endotoxin level in dialysate lower than 0.001 EU/mL) increased from 43.1% in 2009 to 74.6% in 2018 [3], and the number of patients treated with online hemodiafiltration increased dramatically from 16,853 (5.8% of all maintenance dialysis patients) in 2009 to 144,686 (42.0%) in 2019 in Japan [4]. All of these improvements may contribute to a better quality of life and a lower mortality rate in dialysis patients in the world, however, the Japanese registry data also show that the crude mortality rate in these patients has been almost the same, approximately 10%, in recent years [4]. The reasons for this phenomenon may vary, but an increase in the elderly population and changes in social behaviors are considered important factors associated with an increase in dialysis patients with lower physical activity and frailty, resulting in lower quality of life and higher mortality. In fact, the proportion of dialysis patients who have been on dialysis for more than 20 years increased from less than 1% in 1992 to 8.4% at the end of 2018 [3,5]. As a result, the percentage of patients with multiple disabilities has also increased. In addition, dialysis-related complications, including malnutrition, dialysis-related amyloidosis, and skeletal joint disabilities, are still unsolved problems that significantly decrease patients’ quality of life [6,7,8,9,10,11,12,13,14]. Many clinical workers in dialysis centers in developed countries may recognize the clinical relevance for patients with multiple disabilities, sarcopenia, joint pain, and fatigue, despite advances in dialysis technologies. One solution to resolving this issue is to extend the healthy life expectancy so that everyone can continue to live healthy and autonomous lives. Efforts are underway in countries all over the world to develop new treatment strategies to extend the healthy life expectancy of CKD patients, including those on dialysis. Here, the history and concepts of renal rehabilitation, the current status of the dialysis population, the implementation of renal rehabilitation in this population, and future perspectives are reviewed.

## 2. History and Conception of Renal Rehabilitation

Previously, rest was considered one of the treatment options for CKD, especially for patients with nephrotic syndrome, because there were reports in the 1990s suggesting that exercise may worsen the level of proteinuria and renal function. As it became clear that exercise-induced proteinuria was temporary and reversible without renal function loss, exercise therapy for CKD patients gradually gained the interest of nephrologists, dialysis health professionals, and rehabilitation therapists. At the same time, there has been an increase in the elderly population and an increase in the rate of frailty in CKD patients in the world. Since frailty is closely associated with higher mortality and lower quality of life, there is an urgent need to clarify the effects of exercise intervention in CKD patients [12,13,15,16]. To address these concerns, the Japanese Society of Renal Rehabilitation (JSRR) was established in 2011. Rehabilitation is defined by the WHO as “all means to alleviate the effects of conditions that may bring about disabilities and social disadvantages and achieve social integration of people with disabilities and social disadvantages.” Renal rehabilitation was defined as “a long-term comprehensive program consisting of exercise therapy, diet therapy and water management, drug therapy, education, psychological/mental support, etc., to alleviate physical/mental effects based on kidney disease and dialysis therapy, prolong the life expectancy, and improve psychosocial and occupational circumstances.” Rehabilitation in its original form means conducting all possible treatments and exhausting all support options to help kidney disease patients smoothly achieve social rehabilitation instead of simply implementing exercise therapy. Because renal rehabilitation is a comprehensive, multidisciplinary concept for patients with CKD, it is essential that the healthcare professionals associated with CKD treatment, including doctors, rehabilitation therapists, diet nutritionists, nursing specialists, social workers, pharmacists, and therapists, collaborate. All of these health professionals are equally essential components of the program [17]. On the November 2016, a group of international researchers and clinicians first met as the Global Renal Exercise (GREX) Working Group to discuss research priorities related to exercise in CKD at Chicago. They are having regular meetings to foster collaborative research and innovations across multiple disciplines to develop effective and feasible strategies to increase physical activity in CKD patients. Nowadays, many study groups in the world are working for renal rehabilitation.

## 3. Guidelines for Renal Rehabilitation

While a number of papers and narrative reviews regarding exercise in CKD have been published in the 2010’s, there was no comprehensive guidelines related to exercise and physical activity for patients with CKD. The Kidney Disease Improving Global Outcomes (KDIGO) 2012 clinical practice guideline and other guidelines recommended to increase physical activity levels [18]. In 2013 and 2014, both the Exercise and Sports Science Australia position statement and the American Colleges of Sports Medicine guideline recommended aerobic, resistance, and flexibility exercises in CKD patients [19,20]. However, at this point, there was no specific suggestions regarding the types, intensity, and volume of specific types of exercises for CKD patients. The JSRR published the “Guide for Renal Rehabilitation for Predialysis Stage Renal Failure” on their website in 2016 to clarify the recommended exercise menus for CKD patients. Moreover, exercise therapy for diabetic patients with CKD were newly approved by the health insurance system of Japan in 2016. 

Through these developments, the clinical practice guidelines for renal rehabilitation by the JSRR were established in 2018 [21]. This was the one of the first sets of clinical practice guidelines for renal rehabilitation based on systematic reviews and body of evidence. Since exercise therapy is the core of a comprehensive renal rehabilitation program, the relatively rich literature about exercise therapy for CKD patients was primarily reviewed. Thereafter, a number of systematic reviews for physical activity, renal function, lifestyle change in CKD including dialysis patients, as well as international surveys and editors’ commentary, has been published in recent years [22,23,24,25,26,27,28,29,30,31,32].

## 4. Physical Activity in Dialysis Patients

Physical activity levels in dialysis patients are drastically reduced compared to the general elderly population, because dialysis patients tend to have sedentary lifestyles on the day of dialysis, probably due to inactivity for the dialysis procedure and postdialysis fatigue syndrome [33,34,35,36]. It has been reported that the physical activity of dialysis patients is 17% lower on dialysis days than on nondialysis days [36]. Dialysis patients are often exposed to several factors associated with decreased physical activity, such as catabolic disorders that may cause loss of muscle mass and lead to sarcopenia [37], mitochondrial dysfunction [38,39], and comorbidities such as anemia [40], bone and mineral disorders [41], protein energy wasting, diabetes, neurological dysfunction [15,42], and cardiovascular dysfunction [43]. A recent study of nursing home residents in the United States showed that the initiation of dialysis was associated with a decline in functional status, independently of age, sex, race, and functional status trajectory before the initiation of dialysis [44]. Consequently, the physical function of elderly dialysis patients is reportedly approximately half that of the general population [45]. The prevalence of frailty in the ESRD population in the United States is very high and is strongly associated with mortality and hospitalization, even after adjustment for well-established risk factors across multiple domains [46]. Frailty is a similar but not identical concept to sarcopenia. Frailty includes age-associated declines in lean body mass, strength, endurance, balance, walking performance, low activity, and physiological disorders, and the diagnostic criteria of the Cardiovascular Health Study (CHS), the FRAIL scale, or criteria for domestic populations such as the Japanese version of the CHS criteria (J-CHS) have often been used in clinical practice [47,48,49]. More importantly, frailty is a concept of reversible physical, cognitive, and/or social disability, which rehabilitation may play an important role to overcome it [50,51].

Physical activity and exercise habits are closely associated with better outcomes. The Dialysis Outcomes and Practice Pattern Study (DOPPS) reported that patients who habitually exercised more than once a week had better outcomes across all DOPPS countries regardless of physical status or social factors [52]. Many observational studies have suggested an association between physical activity and lower mortality in both CKD and dialysis patients [13,26,53,54]. Matsuzawa et al. also reported that dialysis patients with physical activity of more than 50 min per day and who took a median of approximately 4000 steps a day had better outcomes [55]. Even in patients with CKD with multiple disabilities, it was reported that a less sedentary lifestyle was associated with better outcomes [56]. Of course, a higher level of activity is desirable not only in terms of dialysis mortality, but also for general health; however, efforts to avoid a sedentary lifestyle are the first and most important step for dialysis patients with frailty.

On the contrary, there is still a large gap between theoretical knowledge and real-world clinical practice. The JRDR reported that more than 60% of dialysis patients of all ages and dialysis vintages did not have an exercise habit. The proportion of patients without an exercise habit increased with older age and longer dialysis vintage. In Japanese patients older than 75 years and with a dialysis vintage of over 40 years, for example, almost 80% of patients had no exercise habit [3]. A recent study indicated that only 6.9% of CKD patients met the recommended physical activity levels [57].

## 5. Exercise Tolerance in CKD Patients

Exercise is not only performed by skeletal muscle contraction. The energy supplied by the circulatory and respiratory systems is needed for skeletal muscle contraction. These interactions between metabolic, circulatory, and respiratory functions are called “Wasserman’s gear” (Figure 1) [58]. This schematic diagram demonstrates the importance of cooperative work among the skeletal muscle, cardiovascular system, respiratory system, and nervous system to achieve exercise tolerance. The intensity of aerobic exercise training is a key issue in renal rehabilitation, since exercise intensity is directly linked to both the amount of improvement in exercise capacity and the risk of adverse events during exercise. Therefore, although assessing muscle function is an essential part of exercise, we need to undertake a comprehensive assessment of these organs to assess exercise tolerance.

Exercise tolerance is defined by oxygen-dependent biological systems, the respiratory system, and the circulatory system, and by oxygen consumption by mitochondria in muscular tissue, as described above. Peak oxygen uptake (VO_2_) and the first and second ventilatory thresholds (i.e., the physiological descriptors of the O_2_ transport and utilization system in response to exercise) are the gold-standard references for the evaluation of aerobic metabolic function and, consequently, for aerobic exercise intensity assessment and prescription [59]. Cardiopulmonary exercise testing (CPX) may be the most widely used method to assess exercise tolerance. The clinician’s guidelines for exercise testing, with a comprehensive overview of CPX, are described in detail in the scientific statement paper from the American Heart Association [60]. Modern CPX systems allow for the analysis of gas exchange at rest, during exercise, and during recovery and yield through breath-by-breath measures of oxygen uptake, carbon dioxide output, and ventilation. As a consequence, they also allow for the analysis of the submaximal index of exercise capacity, called the anaerobic threshold (AT) or ventilatory threshold (VT). These parameters are often measured before initiating renal rehabilitation to achieve treatment goals efficiently and safely.

As described above, most CKD patients have very low exercise tolerance because of a decrease in muscle mass due to CKD-related catabolic status, mitochondrial dysfunction, cardiovascular complications, CKD-related mineral bone disorders, and anemia. Exercise tolerance is a strong prognostic factor associated with mortality independent of renal function. In fact, it has been reported in patients with renal transplantation that recovery from physical dysfunction and cardiovascular risks is limited after improvement of renal function [61]. Therefore, increasing or maintaining exercise tolerance is a key factor to improve quality of life in CKD patients.

## 6. Muscle Energy Metabolism and Nutrients

### 6.1. Mechanisms of Adenosine Triphosphate (ATP) Production

Skeletal muscle contraction is a movement produced by the free energy extracted by the hydrolysis of adenosine triphosphate (ATP). ATP is produced in the muscle anaerobically and aerobically. The earliest mechanism for ATP production in the muscle is the creatine phosphate pathway, which utilizes the creatine phosphate (PCr) stored in skeletal muscle (Figure 2). This is a kind of reserve mechanism for high-intensity exercise, where energy demand rises rapidly. However, performance is usually maintained for only approximately 10 s. Thereafter, another anaerobic ATP production system, called the glycolysis pathway, produces 2ATP from glycogen, whose metabolic capacity is about one-half to one-seventh that of the PCr pathway. Then, the main body of ATP production is gradually switched from anaerobic to aerobic exercise, which can produce ATP under oxygen supply. Therefore, organs in the Wasserman’s gear are very important players in aerobic exercise. In order to produce ATP aerobically, energy sources, including sugars, lipids, and proteins, are needed. Sugars and lipids are preferentially used for ATP production. Thereafter, most of the ATP for exercise is produced through the tricarboxylic acid cycle (TCA cycle or Krebs cycle) and the electron transport chain [62].

### 6.2. Synthesis of Skeletal Muscle and Its Related Nutrients

Muscle is the most important and largest site of storage of amino acids in the body. After digestion, carbohydrates are broken into glycogen, proteins into amino acids, and lipids into fatty acids. Glycogens and muscle protein formed by amino acids are stored in the muscle. When glycogen is used up and/or malnutrition is present, muscle protein is broken down into amino acids. When performing exercise therapy, it is necessary to supply additional energy to replace the energy burned by exercise, based on METs (metabolic equivalents). The amount of replenishment energy is calculated as 1.05 × METs × time × body weight. In addition, because elderly people have a higher amino acid threshold for anabolic and lower gastrointestinal absorption capacity, more protein is needed to maintain the nitrogen balance. 

Ingestion of branched-chain amino acids (BCAAs) such as valine, leucine, and isoleucine, especially leucine and its metabolite 3-hydroxy-3-methylbutanoic acid, is important to promote the synthesis of skeletal muscle protein [63]. BCAAs are abundant in dairy products and animal proteins. 

Maintaining high blood amino acid levels after exercise is important for skeletal muscle maintenance and enhancement. BCAA-rich whey protein can relatively rapidly increase the level of blood amino acid concentration after ingestion, while casein and soy protein have a relatively slow digestion and absorption rate. Approximately 20% of milk protein consists of whey protein and 80% of that consists of casein. Therefore, milk protein is expected to both promote and sustain muscle protein synthesis. A recent meta-analysis reported that glycogen synthesis rates are enhanced when carbohydrates and protein are co-ingested after exercise compared to carbohydrates only, when the added energy of protein is consumed in addition to, not in place of, carbohydrates, suggesting the importance of an increase in the energy intake [64]. It is well known that modulating postexercise nutrition is an effective approach to enhance the replenishment of muscle glycogen stores. Considering the absorption time, it may be preferable to ingest protein and carbohydrates before exercise, or just after exercise in the case of fast-absorbing nutrients such as milk protein or amino acids. For athletes, it is recommended to ingest 1.2 g/kg of carbohydrates per hour for 4–6 h postexercise and 0.3 g/kg of a high-quality protein to stimulate muscle protein synthesis and repair [65]. This recommendation for athletes may not be exactly suitable for ESRD patients; however, we need to keep in mind that nutritional support is important for improving muscle functions and physiological performance in ESRD patients.

### 6.3. Muscle Energy Metabolism and Nutrients in CKD

Skeletal muscle atrophy in renal dysfunction begins in the early stages of CKD and progresses with the progression of renal impairment [66]. Skeletal muscle atrophy is caused by an imbalance between protein synthesis and degradation, but there are various causes of skeletal muscle loss in CKD patients [67]. It was reported that the pathophysiology of muscle wasting and weakness is complex and multifactorial. One or more of (1) insufficient nutritional intake; (2) catabolic effects of dialysis therapy; (3) hormonal abnormalities of anabolic hormones (e.g., testosterone, growth hormone, insulin-like growth factor-1), catabolic hormones (e.g., cortisol), or thyroid hormone; (4) chronic inflammation; (5) metabolic acidosis; and (6) concurrent comorbidities [68]. Patients with CKD have a high prevalence of protein energy malnutrition. In order to avoid extra release of muscle protein and to retain muscle in the body, it is essential to meet one’s energy requirements to keep nitrogen balance in CKD patients. For dialysis patients, a 30–35 kcal/kg/day energy intake is recommended, because the estimated energy leak during 4 h of dialysis is approximately 300 kcal. In addition, during the dialysis session, there is a 6–13 g amino acid loss, as well as albumin losses by dialysis circuit absorption and by leakage from the dialysis membrane. Therefore, it is usually recommended to ingest 1.0 to 1.2 g/kg/day of protein, which is 1.2 times higher than recommended for healthy individuals. Furthermore, elderly people should increase their protein intake by 20%. 

In addition to protein and energy intake, sufficient intake of vitamins and minerals such as vitamin D and iron is also very important to prevent muscle protein catabolism [69]. Vitamin D deficiency is common phenomenon in CKD patients. It is reported that maintaining appropriate vitamin D levels is crucial to protect muscle from significant atrophy [70]. Moreover, iron deficiency in skeletal muscle metabolism is strongly associated with lower exercise tolerance independent from anemia, since it is an essential component of oxygen uptake, transport, storage, erythropoiesis, the mitochondrial electron transport chain, and antioxidant enzymes [71]. Uremic toxins such as indoxyl sulfate induce oxidative stress and reduce exercise tolerance and mitochondrial function [72]. There is a report suggesting that reducing the level of indoxyl sulfate results in improvement of exercise tolerance and skeletal muscle function [73]. It has also been reported that EPA and DHA have the effect of suppressing skeletal muscle decomposition [74]. Milk contains a lot of vitamin D, which is necessary for maintaining skeletal muscle. Though it is a very promising ingredient, proper serum phosphorus concentration control using a phosphorus binder is also important. It was reported in the USA that higher protein intake and a concurrent decline in serum phosphorus appear to be associated with the lowest mortality, and diligent use of potent phosphorus binders may be helpful in the dialysis population [75]. The risk of controlling serum phosphorus by restricting dietary protein may outweigh the benefits of controlled phosphorus and may lead to greater mortality. 

## 7. Efficacy of Exercise in Dialysis Patients

With the increase in studies and review papers suggesting the benefits of physical exercise for dialysis patients [76,77,78], scientific societies started to recommend exercise therapy for dialysis patients in the 2010s. In 2012, the KDIGO clinical practice guidelines for CKD encouraged participants to undertake physical activity compatible with cardiovascular health and tolerance, aiming for at least 30 min, five times per week [18]. In 2013, Exercise & Sports Science Australia issued a position statement concerning exercise therapy for CKD patients that describes specific methods of exercise therapy for patients with end-stage kidney disease, both during dialysis and on nondialysis days [19]. It recommends up to 180 min of aerobic exercise with an intensity of 11–13 on the rating of perceived exertion (RPE) scale, 8–12 sessions of resistance exercises with a 60–70% repetition maximum on two nonconsecutive days per week, and 10 min of flexibility exercise 5–7 days per week for dialysis patients. In 2014, the American College of Sports Medicine released its guideline for exercise testing and prescription [20], and specific methods and cautions about exercise therapy for dialysis patients are presented in the latest edition. These guidelines in the 2010s were the standards for exercise therapy in the dialysis population. However, most of these recommendations were based on observational studies, and it was unclear whether exercise intervention was effective for dialysis patients with frailty, until 2018. In 2018, what was, to the best of my knowledge, the first set of evidence-based clinical guidelines for exercise therapy in dialysis patients was published [21,79].

In this systematic review, survival, exercise tolerance, QOL, physical ability (walking ability), physical function (muscle strength), muscle mass, albumin, activity of daily living, Kt/V (a marker of dialysis adequacy), and C-reactive protein (CRP) were selected as outcomes. A meta-analysis of 41 randomized controlled trials demonstrated the significant efficacy of exercise therapy on exercise tolerance (mean difference (MD) in VO_2_: 5.25 L/min/kg, 95% confidence interval (CI): 4.30–6.20 L/min/kg), QOL (MD of physical component summary: 7.39, 95% CI: 2.26–12.51; MD of mental component summary: 9.46, 95% CI: 0.26–18.65), physical ability (MD of 6 min walking distance: 30.2 m, 95% CI: 24.22–36.07 m), and Kt/V (MD: 0.07, 95% CI: 0.01–0.14) (Figure 3, Figure 4, Figure 5 and Figure 6) [21]. Similar results were found in another systematic review [23]. However, no statistically significant difference was noted in terms of muscle strength, muscle mass, albumin, or CRP, although they were all improved. In addition, no significant improvement in survival was observed, perhaps due to the small number of events.

As for the duration and type of exercise, it has been reported that more than six months of intradialytic exercise intervention has significant effects on exercise tolerance (VO_2_) [80]. The same meta-analysis showed that the improvement in exercise tolerance was greater in clinical studies using both aerobic exercise therapy and resistance training than in studies using aerobic exercise therapy alone. There has also been a report showing that the improvement in exercise tolerance is greater with exercise therapy under supervision on nondialysis days, despite a larger number of dropouts [81]. In conducting exercise therapy, the relationship between such specific methods and the effectiveness of exercise therapy must be considered. It is highly significant that physical ability and QOL were improved by exercise, as these are two major components of renal rehabilitation. In this respect, renal rehabilitation plays a major role in achieving the goals of dialysis therapy. In the future, it will be necessary to validate the optimal method of exercise therapy for peritoneal dialysis patients and its effectiveness.

In addition, a recent study suggested that exercise may increase the levels of nitric oxide and myokines and decrease the level of activated oxygen, which are associated with improvements in capillary function, insulin resistance, and other aging-related factors [82,83].

## 8. Implementation of Exercise in Dialysis Patients

The target of exercise therapy is patients with a stable physical condition. The three important steps for exercise therapy in dialysis patients are prior physical evaluation, prescription of an adequate exercise menu, and provision of a continuous support program. Since dialysis patients often have cardiovascular complications, it is essential to promptly assess before prescription whether the patient’s cardiovascular functions and laboratory data, including serum potassium and anemia, are amenable to exercise therapy. In diabetic patients, diabetic complications, including retinopathy, neuropathy, and diabetic foot, may affect the target levels of exercise. If some vital changes are observed, it is preferable to stop exercise until the problems are resolved.

### 8.1. Prior Evaluation

The guidelines for rehabilitation in patients with cardiovascular disease by the Japanese Circulation Society state that, prior to beginning exercise training programs, participants should be assessed for clinical status and should undergo examinations at rest and exercise stress tests to determine the appropriateness of exercise training and to establish the appropriate exercise prescriptions [84]. Acute or uncontrolled cardiovascular diseases are contraindications for exercise. Those with a blood pressure ≥180/100 mmHg, a fasting blood glucose level ≥250 mg/dL, and a body mass index ≥30.0 are also considered contraindications for exercise. In addition, the goals of rehabilitation should be modified based on the patient’s condition. See details in these guidelines or the JSRR renal rehabilitation guidelines.

### 8.2. Four Components of an Exercise Intervention

There are four main components in an exercise intervention: aerobic exercise, represented by walking and swimming; resistance training, represented by push-ups and squats; flexibility exercise, represented by stretching; balance training, represented by standing on one leg (Figure 7). The examples of four types of exercise can be seen in the National Institute on Aging homepage [85]. At this point, it is still unclear which combination is the best for dialysis patients. Most previous papers chose a combination of aerobic and resistance training with some flexibility training. Recent studies of patients with severe renal impairment comparing the effectiveness of resistance training and balance training concluded that the beneficial effects on physical activity and renal function were similar in both groups [86,87]. Therefore, it is preferable to combine these components in a balanced way or to modify them individually to maximize the treatment effects of exercise in dialysis patients.

### 8.3. Menu of Exercise Intervention

The JSRR guidelines recommend that dialysis patients evaluate the causes of walking disability if they have a decreased comfortable walking speed (<1.0 m/s) or short physical performance battery (SPPB) (<12 points) [79]. If patients do not have a walking disability, the first target goal is more than 4000 steps on nondialysis days and walking more than 30 min more than five days per week. In addition, evaluations should be done every six or 12 months. In patients with physical disability, lower-grade center exercise with supervisors should be undertaken.

The exercise prescription should be based on the concept of “FITT”: frequency, intensity, time, and type of exercise. On this point, an example of a suitable exercise program for dialysis patients is a combination of aerobic exercise (e.g., walking ≥ 30 min, five times per week), resistance training (e.g., Thera-band exercise for 10–20 min with a rating of perceived exertion (RPE) of 13–17), and balance training (e.g., double-/single-leg balancing for 5 min 3–5 times a week) [79]. As described before, the rate of frailty in dialysis patients is very high. Therefore, it is preferable to start with a low intensity and adjust it based on the patient’s physical condition.

## 9. Barriers to CKD-Specific Exercise Behavior

Why are dialysis patients sedentary? There are barriers to exercise that keep real clinical practice from adhering to the evidence. In other words, in addition to exercise prescriptions, continuous support programs are essential to achieve our goals, especially in the dialysis population. What are the factors associated with behavioral change? Everyone hesitates over starting exercise because of bad weather, no time, not feeling like it, etc. A study from the U.K. suggested some key factors associated with behavioral changes in CKD patients. They reported that barriers to exercise included physical factors (frailty, anemia, and aging), mental factors (fear of injury or aggravating their condition), absence of motivational support (familial support, encouraging, enjoyment, adequate goals, and accomplishment), and environmental factors (supervisors, facilities, and weather) [88]. Poor physical condition as a result of both comorbid conditions and CKD-related symptoms (fatigue, joint pain, and shortness of breath) [89] was felt by the participants to be the predominant barrier to exercise. The perceived psychological barriers to exercise included fear of injury. Some concerns about exercise may be partly due to the lack of information patients receive about the benefits of exercise from healthcare professionals [57]. Patients expressed a need for tailored advice and support from their healthcare professionals regarding the specific exercises that are safe and appropriate for renal patients.

The Standardized Outcomes in Nephrology—Hemodialysis (SONG-HD) study group reported that the core outcomes that are critically important for dialysis patients and health professionals to avoid are fatigue, cardiovascular disease, vascular access, and mortality [90]. Additionally, a patient-reported outcome (PRO) study from Canada reported that the major barriers for the remaining patients were fatigue (55%), shortness of breath (50%), and weakness (49%). If the patients were going to exercise, they wanted to exercise at home (73%) using a combination of aerobic and resistance training (41%), regardless of modality or age category [91]. While most trials have reported beneficial effects on biochemical parameters and possible benefits in terms of reduced mortality, these PRO studies suggest that these promising results are less important for dialysis patients and may not motivate them to stick to an exercise regimen. Rather, they are more interested in how to relieve fatigue and how to recover energy for their daily lives. Therefore, it is essential for us to understand these barriers and to provide continuous support programs to achieve patient satisfaction. In order to keep exercising regularly and actively for a long time, it may be necessary to prescribe a customized “My pace” program, an easy “Accessible” program, and an enjoyable “Together” program: A combined concept of “MAT: My pace, Accessible, and Together”.

## 10. Conclusions

Dialysis-associated health professionals often act as first-line healthcare providers, not unlike home doctors, for dialysis patients. Therefore, they may be considered the best gatekeepers of renal rehabilitation who need to contact other healthcare professionals, such as rehabilitation therapists, diet nutritionists, nursing specialists, social workers, pharmacists, and therapists, to coordinate the patient’s renal rehabilitation. Renal rehabilitation is a relatively new concept. Based on the urgent needs of an elderly society, especially in advanced countries, it is necessary to combine our multidisciplinary knowledge, gather new evidence, and create a sustainable environment for renal rehabilitation.

## Figures and Tables

**Figure 1 nutrients-13-01444-f001:**
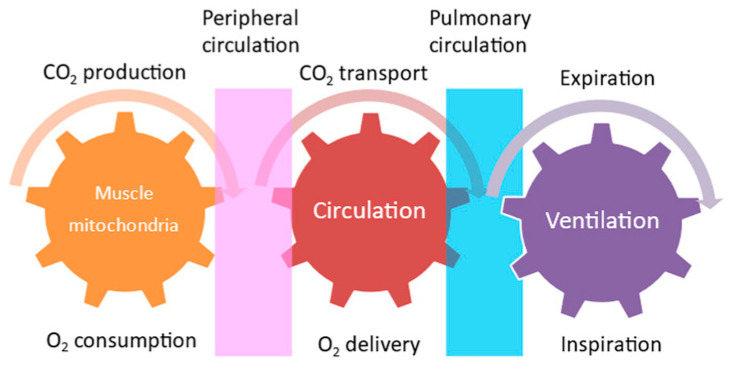
Wasserman’s gear.

**Figure 2 nutrients-13-01444-f002:**
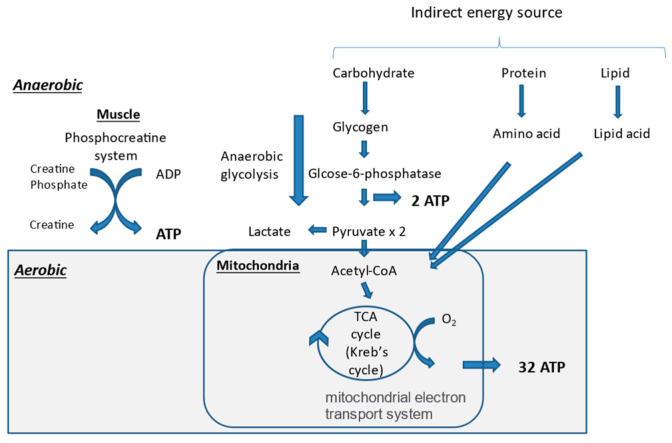
Mechanisms of adenosine triphosphate (ATP) production in muscle. ADP, adenosine diphosphate; TCA, tricarboxylic acid.

**Figure 3 nutrients-13-01444-f003:**
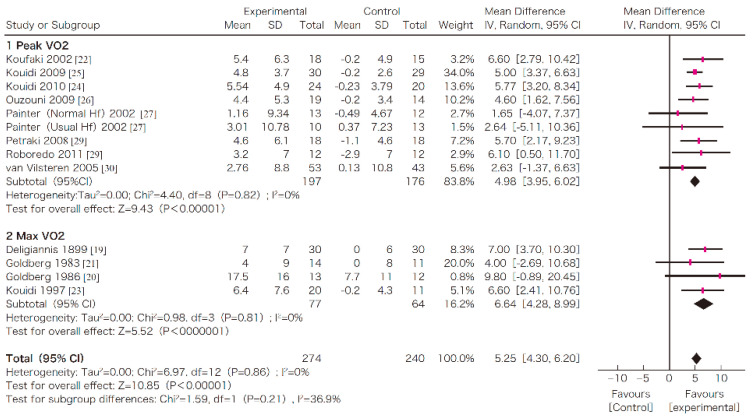
The effects of the exercise intervention on the changes in the exercise tolerance (VO_2_ peak) of dialysis patients [21,79]. SD, standard deviation; CI, confidential interval; IV, inverse variance; VO2, VO_2_.

**Figure 4 nutrients-13-01444-f004:**
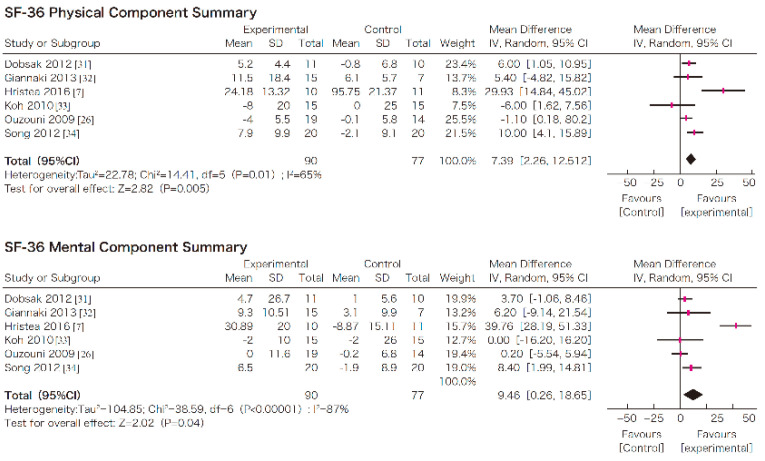
The effects of the exercise intervention on the changes in the quality of life of dialysis patients [21,79]. SD, standard deviation; CI, confidential interval; IV, inverse variance.

**Figure 5 nutrients-13-01444-f005:**
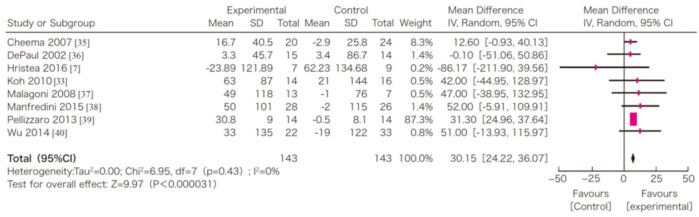
The effects of the exercise intervention on the changes in the 6 min walking distance of dialysis patients [21,79]. SD, standard deviation; CI, confidential interval; IV, inverse variance.

**Figure 6 nutrients-13-01444-f006:**
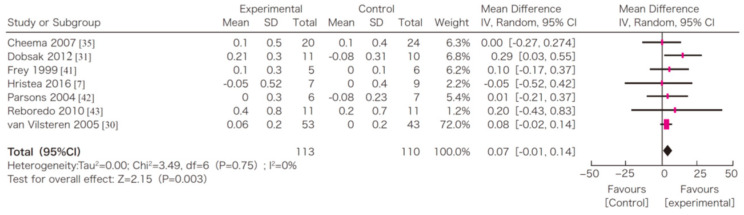
The effects of the exercise intervention on the changes in the 6 min walking distance of dialysis patients [21,79]. SD, standard deviation; CI, confidential interval; IV, inverse variance.

**Figure 7 nutrients-13-01444-f007:**
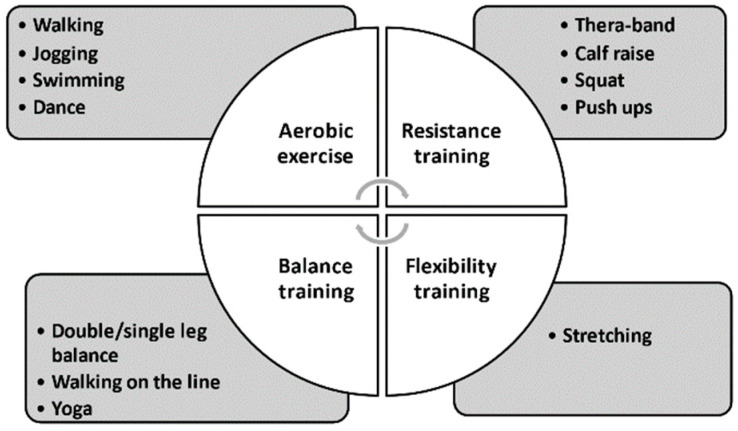
Components in an exercise intervention.

## Data Availability

The data that support the findings of this study are available from the corresponding author, J.H., upon reasonable request. Restrictions apply to the availability of these data. The Figure 3, Figure 4, Figure 5 and Figure 6 was obtained from the references [21,79] and are available from the authors or publishers with the permission.

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
