# Peer review of "Renal Rehabilitation: Exercise Intervention and Nutritional Support in Dialysis Patients"

_nutrients, 2021, doi:10.3390/nu13051444_

Round 1
Reviewer 1 Report
The author provided a narrative review on the effects of exercise and nutritional intervention on patients under dialysis. The term renal rehabilitation was used to describe this process. Overall, the content is not without interest. However, several vital comments are listed below.
- The author claimed that the guideline on renal rehabilitation was the first to be announced in the world (page 3, line 100-101). However, this seems not true since interventions with very similar construct are frequently attempted as reported in the literature, such as intra-dialytic or inter-dialytic exercise, other dedicated exercise programs for dialysis patients, etc. This claim is bold and not perceived to be true, since many recommendations related to exercise encouragement and assessment are available as subparts in KDIGO, KDOQI, or country-specific guidelines on CKD/dialysis patient care.
- Figure 1 is redundant and should be described in words only. Too many words exist in this figure and renders it inappropriate as a figure.
- The first two paragraphs in section 4 (line 118 and afterward) are unrelated to the main theme, physical activity in dialysis patients, and should be substantially condensed and abbreviated.
- Figure 2 is inappropriately constructed. Frailty and sarcopenia are common scenarios in patients with CKD, and the quality of life and mortality is similarly high in these patients. Then why did the authors include only the upper part of this figure in the circle “CKD”? Moreover, the content in this figure is way too simplistic if the authors want to concisely describe the vicious circle of frailty/sarcopenia in CKD patients; if the authors wish to be simple about their message, there is no need to admix molecular process (ex. mitochondrial dysfunction) in the figure, since this pathogenic process occurs in other organs as well.
- The title of Section 5, exercise tolerance in dialysis patients, is narrow. The content in this section includes all CKD patients not just those under dialysis, but the section title is restrictive.
- The content of sections 6.1 and 6.2, muscle energy metabolism and nutrients, is on the contrary too broad and should be corrected. For a review focus on CKD, most content/information/study findings are expected to be derived from cells/animals/humans treated with uremic milieu or with CKD. However, the reports described in this study are quite general and are not specific to those with CKD. It is recommended that the author completely rewrites this section and bases his/her descriptions on CKD-related reports. Alternatively, please condense these sections and merge content with section 7.
- Figures 5 to 8, seems to come directly from the position paper of the author country’s society without reconstruction or redrawing. Wouldn’t this practice violate the copyright requirement of the original reference journal? Copyright issues are serious and should be addressed very carefully.
- Please provide the rationale and the references, published work findings that support the inclusion of relevant exercise types in Figure 9.
Reviewer 2 Report
The article concerns very important problem in elderly patients treated with dialysis. The concept of renal rehabilitation including physical activity and nutritional support is very important.
My detailed comments:
The presented data in the introduction (1) and following parts (line 160-170) come only from Japan and the data from other parts of the world should be added.
Round 2
Reviewer 1 Report
Thank you for responding to the previous comments. I have no further comments.
This manuscript is a resubmission of an earlier submission. The following is a list of the peer review reports and author responses from that submission.